# Transcriptome and Metabolome Profiling Provide New Insights into Disuse Muscle Atrophy in Chicken: The Potential Role of Fast-Twitch Muscle Fibers

**DOI:** 10.3390/ijms25063516

**Published:** 2024-03-20

**Authors:** Zipei Yao, Lijin Guo, Li Zhang, Qinghua Nie

**Affiliations:** National-Local Joint Engineering Research Center for Livestock Breeding & Guangdong Provincial Key Lab of Agro-Animal Genomics and Molecular Breeding & Key Laboratory of Chicken Genetics, Breeding and Reproduction, Ministry of Agriculture, Department of Animal Genetics, Breeding and Reproduction, College of Animal Science, South China Agricultural University, Guangzhou 510642, China; datuohai@stu.scau.edu.cn (Z.Y.); guolijin2016@163.com (L.G.); z15001635024@163.com (L.Z.)

**Keywords:** chicken, disuse muscle atrophy, muscle fiber type, transcriptome, metabolome, energy metabolome

## Abstract

Disuse muscle atrophy is a disease caused by restricted activity, affecting human health and animal protein quality. While extensive research on its mechanism has been studied in mammals, comparatively little is known about this process in chickens, which are a significant source of protein for human consumption worldwide. Understanding the mechanisms underlying skeletal muscle atrophy in chickens is crucial for improving poultry health and productivity, as well as for developing strategies to mitigate muscle loss. In this study, two groups of chickens were subjected to limb immobilization for two and four weeks, respectively, in order to induce disuse muscle atrophy and uniformly sampled gastrocnemius muscle at the fourth week. A combined analysis of the transcriptome and metabolome was conducted to investigate the mechanisms of disuse-induced muscle atrophy. Through H&E staining and immunofluorescence, we found that, compared to slow-twitch muscle fibers, the fast-twitch muscle fibers showed a greater reduction in cross-sectional area in the immobilized leg, and were also the main driver of changes in cross-sectional area observed in the non-immobilized leg. Integrated analysis revealed that differentially expressed genes (DEGs) and differentially accumulated metabolites (DAMs) were mainly enriched in pathways related to energy metabolism, such as fatty acid metabolism, oxidative phosphorylation (OXPHOS), and glycolysis. These results provide important insights for further research on disuse muscle atrophy.

## 1. Introduction

Skeletal muscle is an important component in animals, accounting for about 40% of the body weight in adult animals and containing over half of the body’s protein [1]. The protein in skeletal muscle is also a significant source of agricultural protein for humans as well as for the economic value assessment of agricultural animals [2]. Skeletal muscle has strong contractile and plastic properties, and it can undergo structural and functional changes in response to appropriate stimulation and training to meet the specific demands and environmental conditions of the organism [3]. However, skeletal muscle can also undergo atrophy under pathological or non-pathological conditions [4]. Investigating disuse muscle atrophy in poultry is not only beneficial for enhancing our understanding of poultry health and farming practices but also for broadening our horizons in the field of muscle biology. Under the modern intensive poultry farming and caging conditions, the motion range of broilers was greatly limited, which may lead to muscle atrophy, and it has significantly affected both the economic aspects of the poultry industry and the overall health and growth of the poultry. However, current research on disuse muscle atrophy in poultry remains limited.

Disuse muscle atrophy refers to the wasting of muscles in terms of strength, mass, and structural morphology due to prolonged inactivity, bed rest, immobilization, and even weightlessness in space [4,5]. It is manifested macroscopically as a decrease in muscle mass and a transition from slow muscle fiber type (type I) to fast muscle fiber type (type II) [6]. Additionally, a reduction in muscle fiber diameter is the most prominent pathological feature of muscle disuse atrophy, with affected muscle cells usually smaller and morphologically distinct from normal muscle cells [7]. On a microscopic level, disuse muscle atrophy is primarily characterized by an imbalance between protein synthesis and degradation, with protein degradation being predominant. When synthesis exceeds degradation, protein content increases, promoting muscle hypertrophy. Conversely, when degradation exceeds synthesis, protein content decreases, leading to muscle atrophy [8]. However, the potential mechanism of disuse muscle atrophy in chickens is not yet well understood.

The occurrence of muscle atrophy is a complex process. Increased oxidative stress, inflammatory responses, and mitochondrial dysfunction are considered upstream factors that can trigger protein degradation through pathways such as the ubiquitin-proteasome system, autophagy-lysosome system, caspase system, and calpain system [9,10,11,12]. Among them, the ubiquitin-proteasome system is the primary mechanism for degrading muscle contractile proteins. Research on the changes in metabolites during muscle growth is abundant. IGF-1 can activate the PI3K-Akt-mTOR signaling pathway by binding to its specific receptor (IGF1R1) to enhance protein synthesis, promote satellite cell proliferation, and differentiation [13]. Additionally, in muscle growth, other metabolites such as androgens, insulin, and hepatocyte growth factor (HGF) have been observed to enhance protein synthesis [14,15,16]. However, there is relatively limited research on the changes in metabolites, especially in the context of muscle atrophy, particularly in poultry. Previous studies have shown that artificial muscle atrophy in humans is often accompanied by oxidative stress responses. For example, in a rat model of immobilization-induced muscle atrophy, significant increases in biomarkers of oxidative stress were observed [17]. Up until the present moment, no studies have indicated the impact of changes in other metabolites in disuse muscle atrophy models on this physiological process.

In this study, we aim to explore the phenotypic, transcriptomic, and metabolomic changes occurring in muscles under disuse conditions from macroscopic to microscopic levels. We have established two-week and four-week models of gastrocnemius muscle atrophy in chickens to investigate the changes in muscle fiber types during the atrophy period. Using an integrated analysis of the transcriptome and metabolome, we have identified the involvement of glycolysis in disuse muscle atrophy and explored its role in muscle fiber type shifting.

## 2. Results

### 2.1. Movement Restriction Mediates Muscle Atrophy

To investigate the effects of disuse on chicken muscles, we established a model of disuse-induced muscle atrophy in chickens. To validate this model, we performed Hematoxylin and eosin (H&E) staining on the gastrocnemius muscle and measured the muscle fiber cross-sectional area in each group. From the histological perspective, we observed the following alternations in the gastrocnemius muscles of both the 2-week immobilization groups and the 4-week immobilization groups (refer to Table 1 for detailed grouping information) after the unilateral limb immobilization. The H&E staining revealed that the muscle bundle area in 2W-L group was significantly smaller compared to the 2W-R group and the control group (including the C-L and C-R groups; refer to Table 1 for detailed grouping information, Figure 1A). Specifically, the average cross-sectional area of the muscle bundle in the 2W-L group is 275.45 ± 9.08 μm^2^, which is significantly smaller than the 407.41 ± 29.24 μm^2^ observed in the 2W-R group and the 484.19 ± 33.67 μm^2^ in the C group (Figure 1B,C). Surprisingly, the muscle bundle cross-sectional area in 4W-R is considerably larger than the C group at 590.76 ± 48.61 μm^2^ (Figure 1B,D). The average muscle fiber cross-sectional area of 4W-L is smaller than both the 4W-R and C groups, measuring 248.45 ± 10.36 μm^2^ (Figure 1B,D). In addition, we also measured the equivalent concentric diameter of the total muscle area and analyzed the trends in diameter size in relation to the muscle area (Appendix A). The distribution of muscle cross-sectional area and the equivalent concentric diameter both showed a trend toward normal distribution (Figure 1E–G).

Furthermore, real-time quantitative PCR (qPCR) results revealed genes associated with muscle atrophy, Muscle Atrophy F-box (*Atrogin-1*) and Muscle RING-finger protein-1 (*MuRF-1*) were upregulated in both 2W-L and 4W-L (Figure 1H,I). These results suggest that movement restriction leads to atrophy of the gastrocnemius muscle in chickens.

### 2.2. Fast-Twitch Fiber Has a More Obvious Response in the Atrophy Model

To investigate whether movement restriction leads to fiber-type transformation in disuse-induced muscle atrophy, we performed immunofluorescent staining (Figure 2A). Unexpectedly, the proportion of fast muscle fiber subtype myosin heavy chain 1B (MYH1B) and slow muscle fiber subtype myosin heavy chain 7 (MYH7) did not show statistically significant variations in quantity between C, 2W-L, and 2W-R, nor be-tween C, 4W-L, and 4W-R (Figure 2B). However, when combining the results of aver-age muscle fiber cross-sectional area and average muscle fiber equivalent circle diameter of 2-week immobilization groups and 4-week immobilization groups, we observed a more pronounced magnitude of change induced by movement restriction in fast-twitch muscles.

As for the fast muscle fiber subtype MYH1B, the average muscle fiber cross-sectional area of MYH1B in 2W-L (300.86 ± 9.48 μm^2^) and 4W-L (265.13 ± 6.44 μm^2^) is significantly smaller than that in C-L (538.12 ± 12.94 μm^2^). Furthermore, the average muscle fiber cross-sectional area of MYH1B in 2W-R (434.31 ± 34.14 μm^2^) is comparatively smaller than that in C-R (500.13 ± 20.64 μm^2^), whereas 4W-R (662.58 ± 54.97 μm^2^) exhibits a significant increase compared to C-R (Figure 2C).

When considering the slow muscle subtype MYH7, the muscle fiber cross-sectional area is significantly reduced compared to C-L (332.56 ± 23.78 μm^2^) in both 2W-L (163.20 ± 22.19 μm^2^) and 4W-L (179.31 ± 34.16 μm^2^), but the extent of reduction is not as pronounced as that observed for MYH1B. Moreover, there is no significant alteration in muscle area observed in 2W-R (269.49 ± 38.91 μm^2^) and 4W-R (316.05 ± 31.33 μm^2^) compared to C-R (347.13 ± 17.35 μm^2^) (Figure 2D).

The trend observed in the equivalent concentric diameter of different muscle fiber types is consistent with the muscle cross-sectional area (Figure 2E,F). Based on our analysis of the area of different muscle fiber types, we found that fast-twitch fibers exhibited more pronounced atrophy in 2W-L and 4W-L groups, while they showed more significant hypertrophy in 4W-R groups. Interestingly, in the unilaterally restricted movement, noticeable muscle atrophy was observed in the immobilized left leg, affecting both fast-twitch and slow-twitch muscle fibers. Conversely, strengthening exercises in the right leg led solely to an increase in fast-twitch muscle fibers, with no discernible impact on slow-twitch muscle fibers.

### 2.3. Transcriptome Revealed Potential DEGs Triggering Enhanced Response of Fast-Twitch Fibers in Muscle Atrophy

To investigate the key genes influencing muscle development during disuse-induced muscle atrophy, we collected 20 samples of gastrocnemius muscle from three groups of chickens: the control group, the 2-week immobilization group, and the 4-week immobilization group. We obtained 130.76 Gb clean reads in total, and each sample reached 5.83 Gb with Q30 quality scores no less than 88.58% (Appendix A). From the statistics of comparison results, the comparison efficiency of the clean reads mapped to the reference genome (GRCg6a) of each sample was between 85.86%–89.43%. (Appendix A).

We considered a fold change (FC) ≥ 1.5 and *p*-value < 0.01 as the criterion for differential expression. In the comparison between 2W-R and 2W-L, we identified 209 DEGs, with 116 upregulated and 93 downregulated genes. Similarly, in the 4W-R and 4W-L comparisons, we found 682 DEGs (328 upregulated and 354 downregulated) (Appendix A; Figure 3A,B). This suggests that significant changes in gene expression occur with the onset of disuse-induced muscle atrophy. To further investigate which DEGs drove these changes during movement restriction, we created Venn diagrams and heatmaps of DEGs from the 2W-R versus 2W-L and 4W-R versus 4W-L comparisons (Figure 3C). The results revealed that 64 DEGs appeared in both comparisons. The heatmap of DEGs showed that among these intersecting genes, some genes related to fat deposition, such as the Solute Carrier Family 16 Member 7 (*SLC16A7*) gene, were downregulated in both 2W-L and 4W-L. Additionally, there were some genes related to mitochondrial function that were upregulated in both 2W-L and 4W-L, such as Cholinergic Receptor Nicotinic Delta Subunit (*CHRND*) and Cholinergic Receptor Nicotinic Gamma Subunit (*CHRNG*).

We conducted qPCR on several genes related to these biological processes, and the results were consistent with our observations. Genes associated with glycolysis, such as Phosphoglycerate Mutase (*PGAM*), Phosphoglycerate Kinase 1 (*PGK1*), Glycogen Phosphorylase L (*PYGL*), Glycosylphosphatidylinositol (*GPI*) and Hexokinase 1 (*HK1*) in 2W-L and 4W-L were significantly upregulated (Figure 4A,B), also, genes related to fatty acids synthesis, such as Fatty Acid Synthase (*FASN*), 3-Oxoacyl-ACP Synthase, Mitochondrial (*OXSM*) were significantly upregulated while the gene related to fatty acids β oxidation (*CPT1*) was significantly downregulated (Figure 4C,D), and genes related to mitochondrial activity Cyclooxygenase-2 (*COX2*), *β-globin* in 2W-L and 2W-R were both downregulated (Figure 4E,F). Principal component analysis (PCA) (Appendix A) showed that the gene expression patterns were different among each comparison.

### 2.4. GO,KEGG Enrichment and PPI in Disuse Muscle Atrophy

To investigate the functions of DEGs in the disuse-induced muscle atrophy model, all DEGs were subjected to Gene ontology (GO) and Kyoto Encyclopedia of Genes and Genomes (KEGG) enrichment analyses. In the GO enrichment analysis, we found that 16,179 DEGs were enriched in 2545 biological processes (BP), 381 cellular components (CC), and 658 molecular functions (MF). To highlight significant pathways involved in disuse-induced muscle atrophy, we focused on the top 20 enriched pathways within each of these three categories of the GO enrichment analysis (Figure 5A,B and Appendix A) (Appendix A). In the comparison of DEGs between 2W-R and 2W-L, the top 20 enriched in biological processes (BP) (*p* < 0.005) are mainly associated with lipid metabolism and muscle growth and development pathways, such as lipid storage, low-density lipoprotein particle clearance, muscle cell cellular homeostasis, and glycerol-3-phosphate metabolic process. On the other hand, in the 4W-R versus 4W-L comparison, DEGs are primarily enriched in pathways related to mitochondrial function, cellular development and differentiation, such as mitochondrial respiratory chain complex I assembly, mitochondrial ATP synthesis coupled proton transport, skeletal muscle cell differentiation, and cell cycle arrest.

Although there are no overlapping enriched pathways among the top 20 BP in both comparisons, it is interesting to note that these pathways are all related to energy metabolism and lipid metabolism. The heatmaps generated for the top 20 BP comparisons (Figure 5C,D) reveal DEGs involved in muscle cell composition and regulation of muscle dynamics, such as Myomesin 2 (*MYOM2*) and Cofilin 2 (*CFL2*), in both comparisons. Additionally, ATP5 family genes, such as ATP Synthase Membrane Subunit E (*ATP5ME*) and ATP Synthase Peripheral Stalk Subunit F6 (*ATP5PF*), are downregulated in both 2W-L and 4W-L. To further understand the functions of DEGs, we performed KEGG enrichment analysis. Figure 5E,F showed the top 20 KEGG enriched pathways in the 2W-R versus 2W-L and 4W-R versus 4W-L comparisons (*p* < 0.065), respectively (Appendix A). Cardiac muscle contraction and fatty acid degradation were pathways that were present in both comparisons. In the fatty acid degradation pathway, the Acyl-CoA Synthetase Long Chain Family Member 1 (*ACSL1*) gene was downregulated in both 2W-L and 4W-L. In addition, Forkhead box O (FOXO) signaling pathways associated with oxidative stress and citrate cycle (TCA cycle) associated with energy were also enriched with DEGs, and OXPHOS is significantly downregulated in 4W-L. The heatmap of the enriched DEGs in the top 20 KEGG pathways is shown in Figure 5G,H. Similar to the DEGs enriched in BP, these genes are mostly related to lipid metabolism and energy metabolism, and exhibit similar trends.

We performed protein-protein interaction (PPI) analysis on the selected DEGs (Figure 6A,B). In the comparison of 2W-R versus 2W-L, some DEGs related to fast-twitch muscle fiber development, such as Myosin Heavy Chain 1A (*MYH1A*) (ENSGALG00000037864) and Myosin Heavy Chain 1E (*MYH1E*) (ENSGALG00000029606), exhibited high degrees of connectivity. In the comparison of 4W-R versus 4W-L, proteins related to cell proliferation and cell growth, such as MYC Proto-Oncogene, BHLH Transcription Factor (*MYC*) (ENSGALG00000033631) and Mesenchymal to epithelial transition factor (*MET*) (ENSGALG00000036883), showed high degrees of connectivity.

### 2.5. Identification of Key DAMs in the Metabolome Potentially Enhances Fast-Twitch Fiber Response in Muscle Atrophy

In order to comprehend the metabolic shifts occurring in muscle atrophy induced by unilateral immobilization, we conducted untargeted metabolomic analysis on 20 samples of gastrocnemius. Through this analysis, a total of 2342 metabolites were identified in the default mode after analyzing the data by OPLS-DA under the criterion of FC ≥ 1 or FC ≤ 0.5, Variable Importance in Projection (VIP) ≥ 1, and *p* < 0.05, 130 DAMs including 67 upregulated and 63 downregulated, were identified in the 2W-R versus 4W-L comparison (Figure 7A), whereas 434 DAMs, including 192 upregulated and 242 downregulated, were identified in the 4W-R versus 4W-L comparison (Figure 7B). Moreover, the heatmaps showed clear variations of DAMs under these two comparisons (Figure 7C,D). Interestingly, the top 20 DAMs in 2W-R versus 2W-L are associated with lipid deposition, such as triglyceride, which is significantly upregulated in 2W-R, while Guanosine triphosphate (GTP) is significantly downregulated in 2W-L (Figure 7E) (Appendix A). In the comparison of 4W-R versus 4W-L, Nicotinamide D-ribonucleotide and GTP are both significantly downregulated in 4W-L (Figure 7F) (Appendix A).

We conducted KEGG enrichment analysis on all the DAMs subsequently. 57 and 89 metabolic pathways were enriched in 2W-R versus 2W-L and 4W-R versus 4W-L, respectively (Appendix A). In both comparisons, we observed enrichment of DAMs in the autophagy—animal, Glycerophospholipid metabolism, and citrate cycle (TCA cycle) pathways.

### 2.6. Integrated Analysis of the Transcriptome and Metabolome

To further analyze the effect of disuse-induced muscle atrophy on chickens, weighted gene co-expression network analysis (WGCNA) on DEGs and DAMs was performed (Figure 8A,B). We visualized the top 30 DEGs/DAMs modules based on the absolute values of the correlation coefficients (CC) in a chord diagram, with a co-occurrence frequency threshold of CCp < 0.0107, to better understand the relationship between DEGs and DAMs (Figure 8C,D) (Appendix A). The DEGs module colored Darkslateblue showed significant correlations with many DAMs modules, such as Megrey60, MEtan, and Meturquoise. Many DEGs in this module were enriched in metabolic pathways, such as the biosynthesis of unsaturated fatty acids, the FOXO signaling pathway, and Glutathione metabolism. In the corresponding metabolite module associated with Gedarkslateblue, many metabolites were products of energy metabolism or fatty acid metabolism, such as Nicotinic acid adenine dinucleotide, NADH, and Glutathione. This further supports the notion of a close association between altered gene expression patterns and metabolic shifts in the context of muscle atrophy.

Many DAMs and DEGs were also enriched in different KEGG pathways, both in the comparison of 2W-R versus 2W-L and 4W-R versus 4W-L. We performed the top 10 KEGG pathways, which contained most DEGs and DAMs (Figure 9A,B, Appendix A) (the sum of DEGs and DAMs is at least greater than or equal to 4), of which include fatty acid metabolism, biosynthesis of amino acids, and purine metabolism. Furthermore, the pathway of autophagy-animal and citrate cycles (TCA cycles), which are related to muscle atrophy and energy metabolism, also enriched many DEGs and DAMs. Venn diagrams showed that DEGs and DAMs shared 23 pathways and five pathways in the comparisons, respectively (Figure 9C,D, Appendix A). Tryptophan metabolism (ko00380) was the only pathway these two comparisons shared.

## 3. Discussion

Disuse atrophy of muscles is common in daily life and can occur due to factors such as immobilization caused by fractures [18], bed rest [19], and even in a weightless environment [20]. Molecular-level muscle atrophy can be observed within 8 days [21]. Chickens are considered an ideal animal model for studying skeletal muscle development, as their developmental anatomy is similar to that of mammals, including humans [22]. Unlike traditional farming methods in the past, intensive farming restricts the range of movement for chickens. With breeding and selection, the weight of chickens and the load on their limbs increase, resulting in some degree of muscle atrophy. We simulated a chicken gastrocnemius muscle atrophy model and used 12 chickens to investigate the histological changes as well as explore the underlying causes of these changes through the transcriptome and metabolome during the disuse period. The integrated analysis of the transcriptome and metabolome has revealed an enrichment of pathways related to energy metabolism and fatty acid metabolism in chickens suffering from disuse muscle atrophy. To the best of our knowledge, this is the first study to explore the impact of disuse muscle atrophy on muscle growth and development in chickens through the integrated analysis of the transcriptome and metabolome. These findings may offer valuable research directions for understanding the specific mechanisms underlying disuse muscle atrophy in chickens.

Through H&E staining and immunofluorescence staining, we observed that both 2W-L and 4W-L showed a significant reduction in muscle cross-sectional area during the immobilization-induced disuse period. Furthermore, after measuring the area of different types of muscle fibers, we found that the magnitude of change in the area of fast-twitch muscle fibers was greater than that of slow-twitch muscle fibers. However, there was no change in the proportion of muscle fiber types. These findings differ somewhat from the study by Mo et al., who observed a significant decrease in the cross-sectional area of the gastrocnemius muscle in both legs of chickens after immobilizing the unilateral gastrocnemius muscle for two weeks [23]. Transcriptome sequencing results revealed that DEGs were enriched in pathways such as mitochondrial ATP synthesis-coupled proton, including *ATP5ME* and *ATP5PF*. Previous studies have suggested the involvement of these genes in ATP synthesis [24]. In the integrated analysis, using WGCNA, we found that NAD and NADH were enriched in 2W-R and 4W-R. Skeletal muscle is composed of a series of muscle fibers generated by myocytes [25]. Muscle fibers are derived from myocytes and are a metabolically active cell type that heavily relies on mitochondrial OXPHOS to produce ATP. Previous studies have shown that immobilization of muscles significantly impacts the quantity and quality of skeletal muscle mitochondria through the induction of inflammatory factors and the promotion of ROS [26], muscle disuse or immobilization can lead to a decrease in the activity of mitochondrial oxidative phosphorylation complexes and a reduction in the synthesis of citrate in the tricarboxylic acid (TCA) cycle [27]. Therefore, in the oxidative stress environment created by disuse, the function of mitochondria in cells is compromised and damaged, which affects cellular energy supply [28]. In addition, the accumulation of ROS caused by disuse can activate protein degradation pathways such as AMPK-mediated UPS and autophagy. This leads to an increase in protein breakdown in muscles, which ultimately results in muscle atrophy [29]. In our transcriptome and qPCR results, we observed significant downregulation of the OXPHOS pathway in the 4W-L. Genes related to glycolysis showed significant upregulation, while genes involved in fatty acid β-oxidation, such as *CPT1*, were significantly downregulated in the 2W-L and 4W-L. Additionally, *ACSL1*, which is involved in fatty acid activation in the fatty acid degradation pathway [30], was downregulated in both 2W-L and 4W-L. It has been reported that upregulation of *ACSL1* expression enhances the level of fatty acid oxidation in mice [31]. These findings suggest that in the state of muscle atrophy, the muscle tissue may be more inclined towards glycolysis for energy production than fatty acid oxidation, or OXPHOS.

During the disuse period, we also observed a decrease in the cross-sectional area of skeletal muscles in 2W-R, even though immobilization did not occur. According to the available literature, there is no specific evidence to suggest a clear relationship between changes in the cross-sectional area of muscles on the immobilized and non-immobilized sides during immobilization-induced disuse atrophy. A study by Glover et al., which involved a 14-day experiment with unilateral knee joint immobilization, reported that the non-immobilized leg did not exhibit significant changes in muscle cross-sectional area compared to the immobilized leg [32]. In our experiment, regarding the changes in the cross-sectional area of the gastrocnemius muscle, we propose several explanations. Prolonged muscle immobilization does not immediately reverse the effects of disuse atrophy through remobilization (RM). Instead, the process of immobilization followed by remobilization (IM-RM) can promote the generation of ROS, activate the NFκB pathway, and subsequently stimulate the secretion of pro-inflammatory factors such as TNF-α and IL-6 [33]. This can contribute to increased oxidative stress in the body [34]. During muscle immobilization, the PI3K/Akt/mTOR pathway, which regulates protein synthesis, is inhibited, resulting in decreased protein synthesis. Additionally, the activation of the ubiquitin-proteasome pathway and autophagy pathway enhances protein degradation, leading to muscle loss [35,36]. Furthermore, during immobilization, reduced stimulation of the PI3K/Akt/mTOR pathway leads to dephosphorylation of FOXO3, thereby enhancing the expression of genes such as *MuRF-1, Atrogin-1*, and *ULK1*, which are involved in protein degradation through autophagy and ubiquitination pathways [37,38,39]. The FOXO signaling pathway plays a critical role in skeletal muscle, and FOXO3 is crucial in the development of immobilization-induced muscle atrophy [40]. These findings are consistent with our observations at the transcriptome level, suggesting that these detrimental cycles occurring during IM-RM may be the underlying causes of changes in the cross-sectional area of 2W-R.

In addition, the change in the quantity of muscle fiber type during the experiment was not significant. Skeletal muscle contains multiple fiber types, including type I, IIa, IIb, and IIx, and under certain conditions, these fiber types can convert into each other [41,42]. In disuse conditions, there are changes in the activity of glycolytic and oxidative enzymes in muscle [43], and a phenomenon observed is the transformation from type I to type II fibers [44]. Various factors affect this conversion, including breed, gender, age, nutrition, and exercise [45,46,47]. Time is also important in promoting transformation between different fiber types. In an experiment where electrical stimulation induced changes in nerve activity to mediate fast-to-slow fiber conversion, it was found that two months of low-frequency electrical stimulation did not cause a difference in the proportion of slow fibers in rat muscle, while the transformation between fast and slow fibers was only observed at four months [48]. This suggests that although molecular changes occur during disuse, these changes are not significant enough to cause a statistically significant change in the proportion of muscle fiber types due to the influence of time. For meat quality in livestock, type I fibers contain more hemoglobin and myoglobin, resulting in a redder color and tenderness and juiciness increasing with higher amounts of type I fibers [47,49,50]. In contrast, type II fibers have a higher glycogen content and activity of glycolytic enzymes, leading to lactic acid buildup and adversely affecting meat flavor [51]. Our findings may be good news for the chicken industry since short-term disuse muscle atrophy does not cause significant changes in the number of fast and slow fibers. Additionally, we have observed that muscle hypertrophy in the non-immobilized leg mainly occurs in fast fibers. Compared to slow-twitch fibers, fast-twitch fibers have higher contractile and power output capabilities but are weaker in terms of endurance and fatigue resistance [52]. This means that fast-twitch fibers rely more on high-energy phosphates for energy supply and are more prone to degradation during long-term disuse and malnutrition periods. The ubiquitin-proteasome pathway and the autophagy pathway play an important role in muscle atrophy, both of which are directly regulated by FOXO3, a member of the FOXO family [53]. The transcriptional coactivator PGC-1α is more abundant in slow fibers compared to fast fibers and is involved in the regulation of FOXO3 [8,54]. PGC-1α can inhibit the binding of FOXO3 to the Atrogin-1 promoter and suppress the transcription of Atrogin-1 [55]. Overexpression of PGC-1α can inhibit muscle atrophy caused by starvation and immobilization [56]. Fast fibers have higher glycolytic enzyme activity and lower levels of PGC-1α, making them more susceptible to muscle atrophy induced by immobilization [57]. Exercise can induce the expression of PGC-1α, protect against muscle atrophy, and promote muscle hypertrophy. In transcriptomic sequencing results, pathways related to glycolysis and energy metabolism in 4W-R were upregulated compared to 4W-L, further confirming our speculation. Furthermore, in the four-week immobilization group, the immobilized leg is unable to move freely, resulting in the non-immobilized leg bearing the majority of the body weight. Progressive loading exercises can promote muscle hypertrophy. Similar studies have demonstrated that resistance training or progressive loading can effectively improve muscle condition in animals with muscle atrophy [58,59,60]. Moreover, the integrated analysis revealed that the tryptophan metabolism pathway was the only intersecting pathway identified in the two groups of chickens subjected to immobilization for different lengths of time. Some studies suggest that the tryptophan metabolism pathway may be related to muscle atrophy. For example, low serum concentrations of tryptophan can cause muscle atrophy in mice [61]. Furthermore, supplementing tryptophan in the diet can alleviate the transformation of pig skeletal muscle fiber from type I to type II [62]. However, further research is needed to elucidate the mechanisms underlying the interaction between tryptophan metabolism and disuse muscle atrophy.

This study of disuse-induced muscle atrophy in chickens can provide insights into the mechanisms underlying muscle wasting and help identify potential therapeutic targets. Chickens are a valuable model organism for studying muscle physiology and pathology due to their relatively simple muscular system and the ability to induce disuse atrophy through immobilization. By studying the molecular and cellular changes that occur during disuse muscle atrophy in chickens, researchers can gain a better understanding of the underlying mechanisms involved. This knowledge can be applied to other animal models and humans to develop new treatments for disuse muscle atrophy and related disorders. In addition, studying disuse muscle atrophy in chickens can have implications for improving poultry production, as muscle growth and development are crucial for meat production and quality. Therefore, investigating disuse muscle atrophy in chickens is highly relevant for both basic scientific research and practical applications in agriculture and medicine.

In summary, this study provides new foundational data for disuse-induced muscle atrophy and suggests that glycolysis may play a significant role in disuse muscle atrophy.

## 4. Materials and Methods

### 4.1. Ethics Statement

The experiment was conducted in accordance with established rules and regulations under the approval of the Animal Protection and Utilization Committee at South China Agricultural University under Approved ID: SCAU#2021F074. The experimental animals used in this study were sourced from Kaiping XuFeng Agriculture and Animal Husbandry Co., Ltd. (Kaiping, China).

### 4.2. Experimental Animals

12 Mahuang chickens (49-day-old broilers, female) with similar body weights, good health, and standard immunization were randomly divided into three groups: C (*n* = 4), which includes the C-L group and C-R group; a group with two weeks of unilateral limb immobilization on the left leg followed by two weeks of free activity (*n* = 4); and a group with four weeks of unilateral limb immobilization on the left leg (*n* = 4). The right leg of all chickens was unrestricted and received no treatment, allowing free movement and serving as a control for comparison (Refer to Table 1 for detailed information). The left legs were immobilized using polymer fiber bandages, restricting the muscle groups around the knee joint from free movement but allowing the chickens to maintain a standing position. All chickens were reared under natural 12-h light-dark cycles, provided with ample food and water, and maintained at an appropriate ambient temperature (24−26 degrees Celsius) until the completion of the four-week period. The experiment was conducted in accordance with established rules and regulations under the approval of the Animal Protection and Utilization Committee at South China Agricultural University under Approved ID: SCAU#2021F074.

### 4.3. Hematoxylin and Eosin Staining (H&E)

The gastrocnemius muscle tissue was collected and underwent dewaxing and hydration. Subsequently, it was rewarmed and immobilized. Then, hematoxylin and eosin staining were performed. Finally, the tissue was dehydrated and sealed.

### 4.4. Immunofluorescence Staining (IF) Experiment

We used a primary antibody targeting Myh (Anti-Fast Myosin Skeletal Heavy chain Rabbit pAb; Abcam; ab172967; 1:500) for the chicken gastrocnemius samples. Following the manufacturer’s instructions, we stained the muscle fibers using the Muscle Fiber Typing Staining Kit (Servicebio, Wuhan, China). Subsequently, a secondary antibody (Anti-MSK1 Rabbit pAb; Servicebio; GB11857-100; 1:500) was applied. We then performed nuclear counterstaining using DAPI (Servicebio, G1012) and captured images using the Nikon Eclipse C1 fluorescent microscope.

### 4.5. RNA-Seq and Data Analysis

The total RNA extraction was performed following the instruction manual of the TRlzol Reagent (Life Technologies, Carlsbad, CA, USA). The concentration and purity of RNA were measured using the NanoDrop 2000 (Thermo Fisher Scientific, Wilmington, DE, USA). RNA integrity was assessed using the RNA Nano 6000 Assay Kit of the Agilent Bioanalyzer 2100 system (Agilent Technologies, Santa Clara, CA, USA). The library construction used the Illumina NovaSeq platform, generating 150 bp paired-end sequences. The raw RNA-seq data reported in this study were submitted to the GSA database with accession number: CRA013067.

Gene Ontology (GO) enrichment analysis is implemented through the clusterProfiler R Package. It can adjust for the bias in gene length in DEGs. We used the KOBAS database [63] and the clusterProfiler R Package software version 4.3.2 to analyze the enrichment of DEGs in Kyoto Encyclopedia of Genes and Genomes (KEGG) pathways. GO terms and KEGG pathways with an adjusted *p*-value < 0.05 are considered significantly enriched with DEGs.

### 4.6. RNA Extraction, cDNA Synthesis, and Quantitative Real-Time PCR

According to the instructions of RNA iso (TaKaRa, Otsu, Japan), total RNA was extracted from the gastrocnemius muscle tissue. The cDNA was synthesized from the RNA using HiScript III All-in-one RT SuperMix Perfect for real-time quantitative PCR (qPCR) (Vazyme, Guangzhou, China). The primer design was performed using the National Center for Biotechnology Information (NCBI) Primer Design Tool. The primer information is listed in (Appendix A). qPCR was performed using the SYBR qPCR Master Mix (Vazyme) following the manufacturer’s instructions to detect the relative expression levels of mRNA. The analysis of qPCR results was based on the calculation of 2^−ΔΔCt^ [64].

### 4.7. Metabolite Extraction and LC-MS/MS Analysis

A non-targeted metabolomics analysis was performed on gastrocnemius muscle samples. All samples were analyzed on their liquid chromatography-mass spectrometry (LC/MS) platform, with 4 biological replicates for each sample group.

The LC/MS system used for metabolomics analysis consisted of the Waters Acquity I-Class PLUS ultra-high-performance liquid chromatography (UHPLC) coupled with the Waters Xevo G2-XS QTOF high-resolution mass spectrometer. The chromatographic conditions employed the Waters Acquity UPLC HSS T3 column (1.8 um, 2.1 × 100 mm) [65]. The composition of mobile phase A was 0.1% formic acid in water, while mobile phase B was 0.1% formic acid in acetonitrile. The injection volume was 1μL.

A high-resolution mass spectrometer can collect primary and secondary mass spectrometry data in MSe mode under the control of the acquisition software (MassLynx V4.2, Waters). In each data acquisition cycle, dual-channel data acquisition can be performed on both low collision energy and high collision energy at the same time. The low collision energy is 2 V, the high collision energy range is 10~40 V, and the scanning frequency is 0.2 s for a mass spectrum. The parameters of the ESI ion source are as follows: Capillary voltage: 2000 V (positive ion mode) or −1500 V (negative ion mode); cone voltage: 30 V; ion source temperature: 150 °C; desolvent gas temperature: 500 °C; backflush gas flow rate: 50 L/h; desolventizing gas flow rate: 800 L/h.

The raw data collected using MassLynx V4.2 is processed by Progenesis QI software V 2.3 for peak extraction, peak alignment, and other data processing operations based on the Progenesis QI software online METLIN database and Biomark’s self-built library for identification, and at the same time, theoretical fragment identification and mass deviation. All are within 100 ppm. After normalizing the original peak area information with the total peak area, the follow-up analysis was performed. Principal component analysis and Spearman correlation analysis were used to judge the repeatability of the samples within the group and the quality control samples. The identified compounds are searched for classification and pathway information in KEGG, the Human Metabolome Database (HMDB), and the lipidmaps databases. According to the grouping information, calculate and compare the difference multiples. A student’s *t*-test was used to calculate the *p*-values of each compound. The R language package ropls was used to perform Orthogonal Partial Least Squares-Discriminant Analysis (OPLS-DA) modeling, and 200 times permutation tests were performed to verify the reliability of the model. The Variable Importance in Projection (VIP) value of the model was calculated using multiple cross-validations. The method of combining the difference multiple, the *p*-value, and the VIP value of the OPLS-DA model was adopted to screen the differential metabolites. The screening criteria are Fold Change (FC) > 1, *p* value < 0.05, and VIP > 1. The difference in metabolites of KEGG pathway enrichment significance were calculated using a hypergeometric distribution test.

### 4.8. Integrated Analysis of the Transcriptome and Metabolome

We performed an integrated analysis of the transcriptome and metabolome. The Venn diagram for DEGs and DAMs was based on a package in R, which provides highly customizable Venn [66]. To study the relationship between DEGs and DAMs in pathways, we first used Fisher’s exact test and Bonferroni correction to identify pathways with an adjusted *p* value ≤ 0.05. Then, we mapped the DEGs and DAMs from the same group onto the KEGG pathway map simultaneously [67].

### 4.9. Statistical Analysis

In this study, experiments were repeated at least four times for reproducibility. Area measurements were analyzed using Image J software V 1.52e. Results were presented as mean ± SEM and analyzed for statistical significance using an unpaired Student’s *t*-test (* *p* < 0.05; ** *p* < 0.01; *** *p* < 0.001). In the comparison among multiple groups, we used the one-way ANOVA test and conducted Tukey post-hoc tests using SPSS 27.0 (SPSS Inc., Chicago, IL, USA). Differences were considered statistically significant when *p* < 0.05. GraphPad Prism 8.3.0 was used to create a histogram and bar graph.

## Figures and Tables

**Figure 1 ijms-25-03516-f001:**
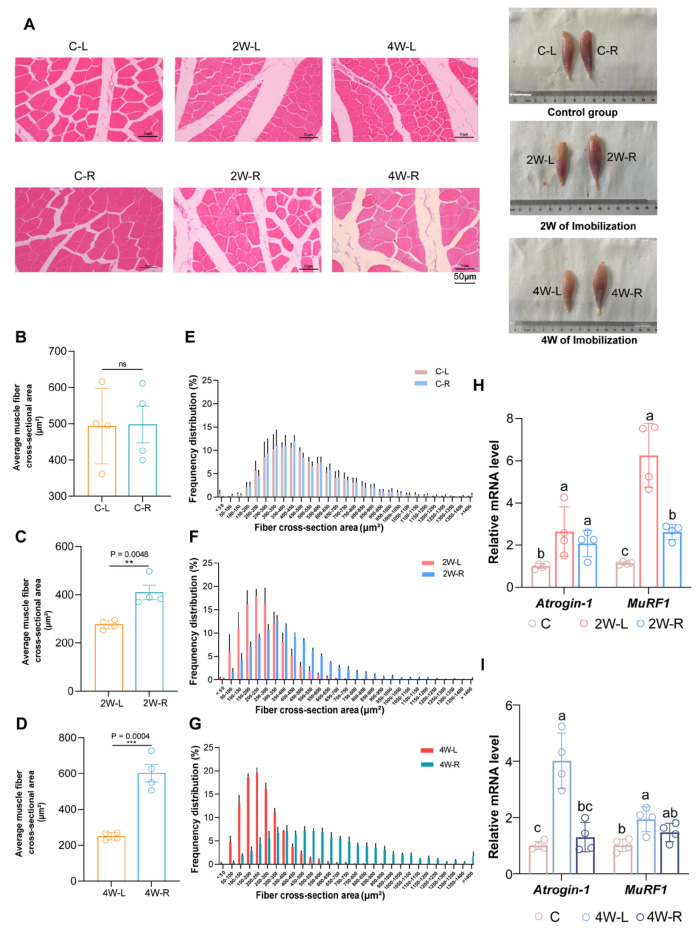
The alternations in the model of muscle atrophy. (**A**) Hematoxylin and eosin (H&E) staining of gastrocnemius muscle (scale bar = 50 µm) and a schematic diagram of the gastrocnemius muscle of each group after sampling. (**B**) Statistics: average muscle fiber cross-sectional area of the C group. (**C**) Statistics: average muscle fiber cross-sectional area of 2W-L and 2W-R. (**D**) Statistics average muscle fiber cross-sectional area of 4W-L and 4W-R. (**E**) Distribution of muscle fiber cross-sectional area of the C group. (**F**) Distribution of muscle fiber cross-sectional area of 2W-L and 2W-R. (**G**) Distribution of muscle fiber cross-sectional area of 4W-L and 4W-R. (**H**) Relative mRNA levels of *Atrogin-1* and *MuRF-1* in C, 2W-L, and 2W-R. (**I**) Relative mRNA levels of *Atrogin-1* and *MuRF-1* in C, 4W-L, and 4W-R. In each group comparison, different letters indicate a significant difference (*p* < 0.05), while the presence of the same letter indicates no significant difference (*p* > 0.05). ns *p* > 0.05; ** *p* < 0.01; *** *p* < 0.001 (Student’s *t*-test).

**Figure 2 ijms-25-03516-f002:**
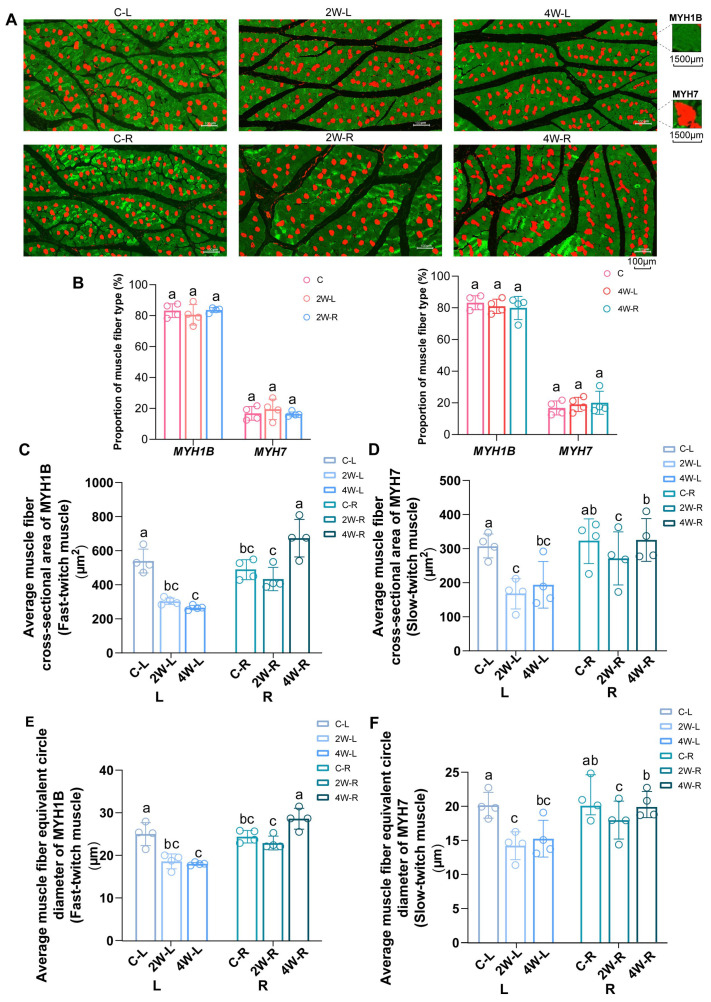
The alternations of muscle fiber composition type in the muscle atrophy model. (**A**) Immunofluorescence staining (IF) of gastrocnemius muscle (scale bar = 100 µm), the color presented by MYH7 is red, while the color presented by MYH1B is green. (**B**) Proportion of different muscle fiber types in C, 2W-L, 2W-R, 4W-L, and 4W-R. (**C**) Average muscle fiber cross-sectional area of MYH1B. (**D**) Average muscle fiber cross-sectional area of MYH7. (**E**) Average muscle fiber equivalent circle diameter of MYH1B. (**F**) Average muscle fiber equivalent circle diameter of MYH7. Data are presented as mean ± SEM (*n* = 4 biologically independent samples). In each group comparison, different letters indicate a significant difference (*p* < 0.05), while the presence of the same letter indicates no significant difference (*p* > 0.05).

**Figure 3 ijms-25-03516-f003:**
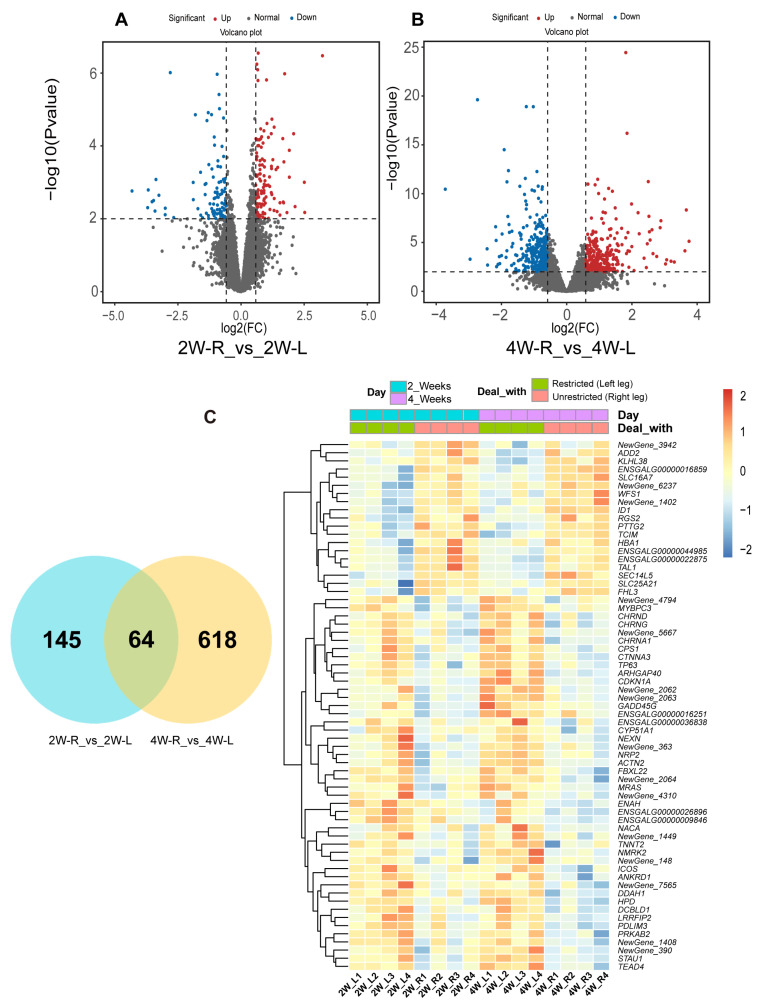
Analysis of DEGs in the muscle atrophy model. (**A**) Volcano plots of DEGs in 2W-R versus 2W-L. (**B**) Volcano plots of DEGs in 4W-R versus 4W-L. (**C**) Venn diagrams and heatmaps of DEGs in the 2W-R versus 2W-L and 4W-R versus 4W-L comparisons.

**Figure 4 ijms-25-03516-f004:**
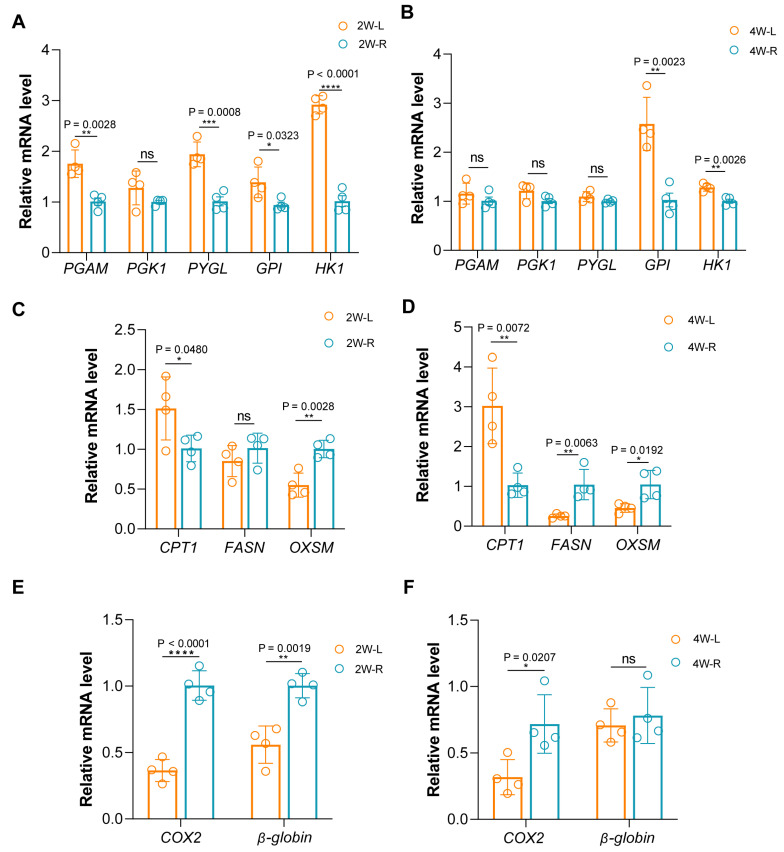
Validation of genes related to muscle atrophy. (**A**,**B**) Relative mRNA level of genes related to glycolysis in 2W-L versus 2W-R and 4W-L versus 4W-R. (**C**,**D**) Relative mRNA level of genes related to fatty acids β oxidation in 2W-L versus 2W-R and 4W-L versus 4W-R. (**E**,**F**) Relative mRNA level of genes related to Mitochondrial activity in 2W-L versus 2W-R and 4W-L versus 4W-R. ns *p* > 0.05; * *p* < 0.05; ** *p* < 0.01; *** *p* < 0.001; **** *p* < 0.0001 (Student’s *t*-test).

**Figure 5 ijms-25-03516-f005:**
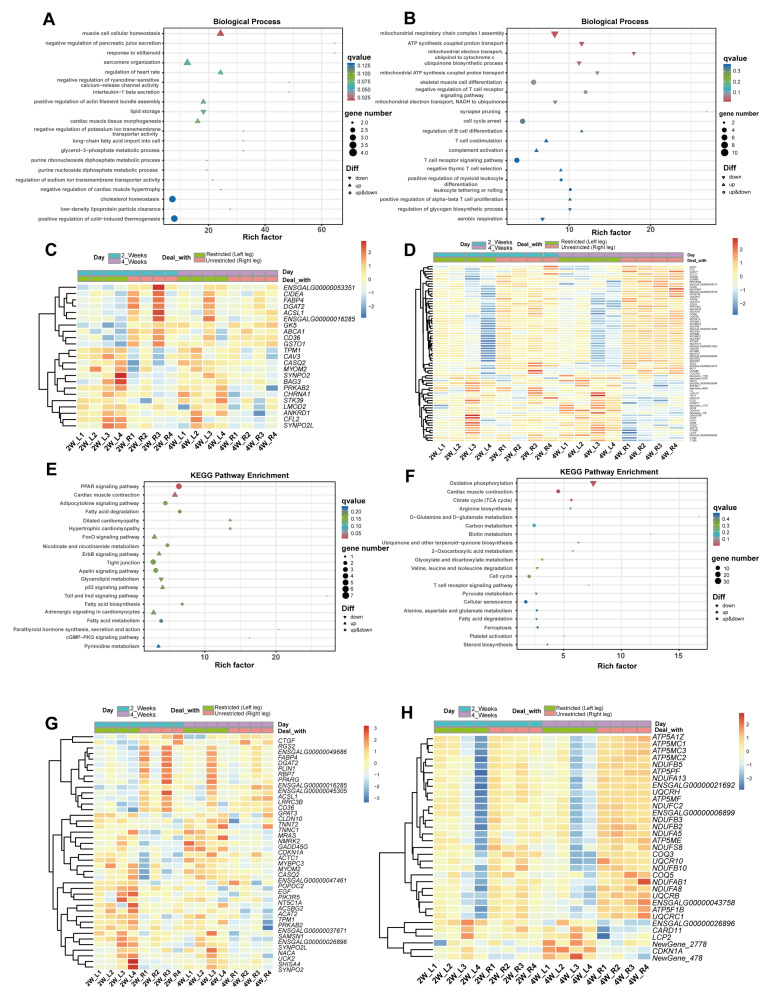
GO and KEGG pathways analysis for DEGs in the muscle atrophy model. (**A**) Top 20 terms of the biological process part of the GO enrichment analysis for DEGs in the comparisons of 2W-R versus 2W-L. (**B**) Top 20 terms of the biological process part of GO enrichment analysis for DEGs in the comparisons of 4W-R versus 4W-L. (**C**) Heatmap of DEGs in the top 20 BP enrichments in the comparisons of 2W-R versus 2W-L. (**D**) Heatmap of DEGs in the top 20 BP enrichments in the comparisons of 4W-R versus 4W-L. (**E**) Top 20 enriched KEGG pathways analysis for DEGs in 2W-R versus 2W-L. (**F**) Top 20 enriched KEGG pathways analysis for DEGs in 4W-R versus 4W-L. (**G**) Heatmap of DEGs in the top 20 KEGG enrichments in the comparisons of 2W-R versus 2W-L. (**H**) Heatmap of DEGs in the top 20 KEGG enrichments in the comparisons of 4W-R versus 4W-L.

**Figure 6 ijms-25-03516-f006:**
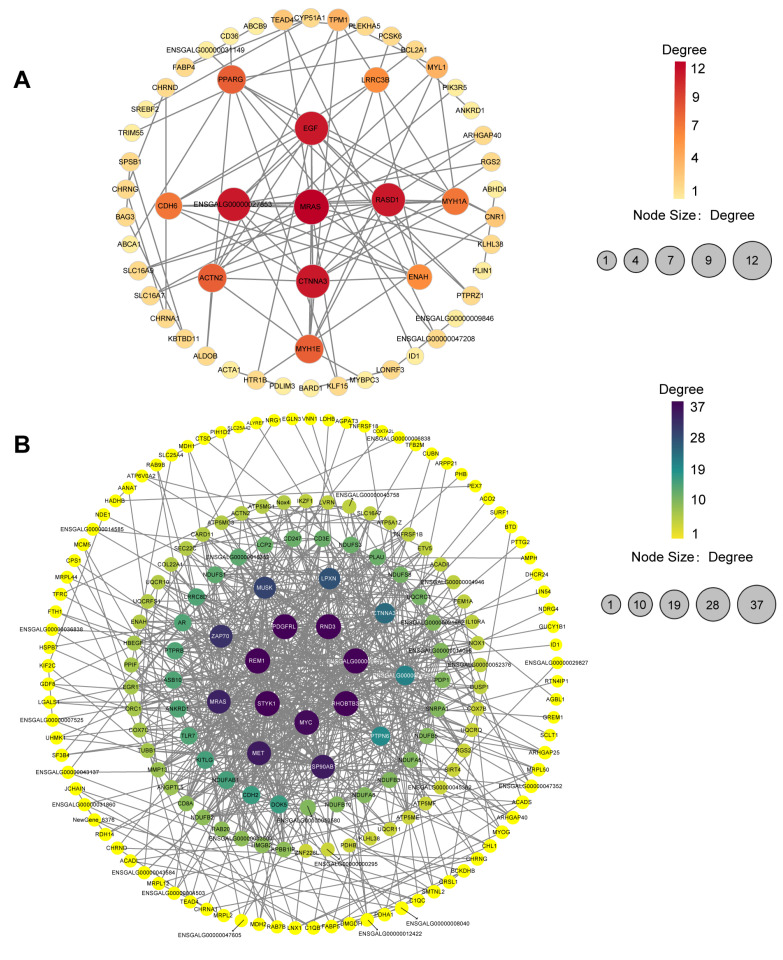
PPI analysis of DEGs in the muscle atrophy model. (**A**) Homologous protein interaction network graph of DEGs in the comparison of 2W-R versus 2W-L. (**B**) Homologous protein interaction network graph of DEGs in the comparison of 4W-R versus 4W-L.

**Figure 7 ijms-25-03516-f007:**
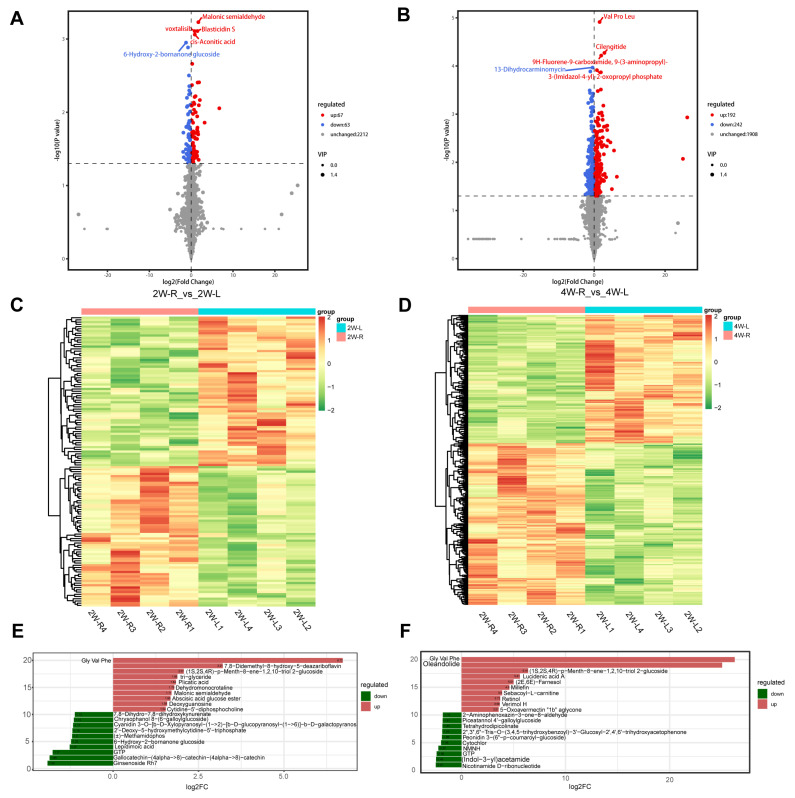
Analysis of DAMs in the muscle atrophy model. (**A**) Volcano plots of DAMs in 2W-R versus 2W-L. (**B**) Volcano plots of DAMs in 4W-R versus 4W-L. (**C**) Heatmap of the DAMs in the comparison of 2W-R versus 2W-L (*n* = 4). (**D**) Heatmap of the DAMs in the comparison of 4W-R versus 4W-L (*n* = 4). (**E**) Top 20 DAMs in 2W-R versus 2W-L. (**F**) Top 20 DAMs in 4W-R versus 4W-L.

**Figure 8 ijms-25-03516-f008:**
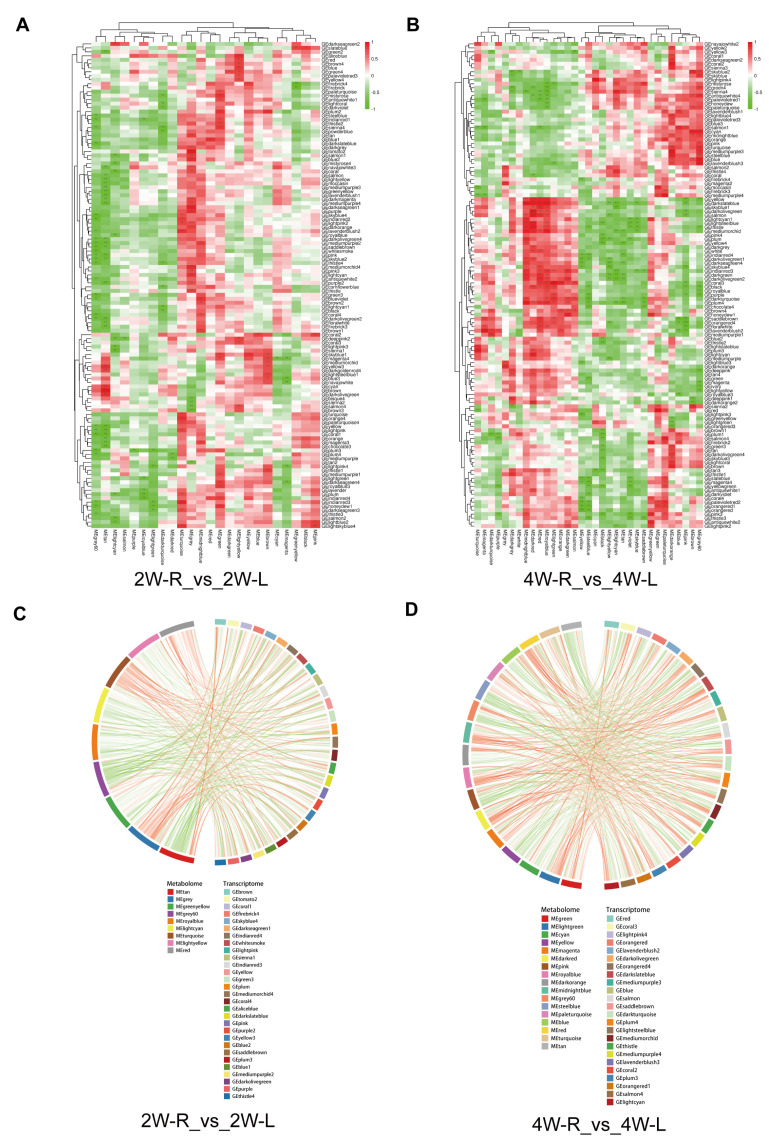
WGCNA of DEGs and DAMs in disuse muscle atrophy. (**A**) Heatmap of correlation for modules with different colors in 2W-R versus 2W-L. (**B**) Heatmap of correlation for modules with different colors in 4W-R versus 4W-L. (**C**) Chord diagram of the top 30 DEGs/DAMs modules in terms of their co-occurrence frequencies in 2W-R versus 2W-L. (**D**) Chord diagram of the top 30 DEGs/DAMs modules in terms of their co-occurrence frequencies in 4W-R versus 4W-L. ** *p* < 0.01; *** *p* < 0.001 (Student’s *t*-test).

**Figure 9 ijms-25-03516-f009:**
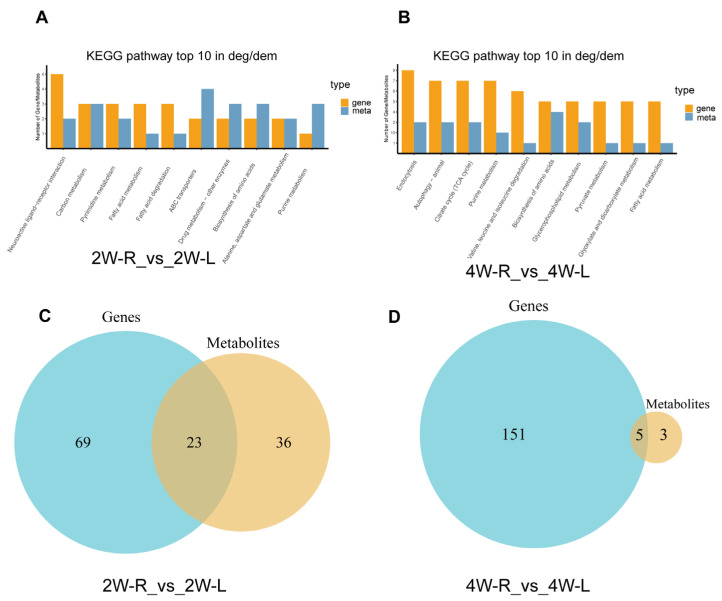
Integrated analysis of DEGs and DAMs in the muscle atrophy model. (**A**) The top 10 pathways with the most DEGs and DAMs in 2W-R versus 2W-L. (**B**) The top 10 pathways with the most DEGs and DAMs in 4W-R versus 4W-L. (**C**) The number of intersecting pathways involved between the pathways in the transcriptome and the pathways in the metabolome in 2W-R versus 2W-L. (**D**) The number of intersecting pathways involved between the pathways in the transcriptome and the pathways in the metabolome in 4W-R versus 4W-L.

**Table 1 ijms-25-03516-t001:** Experimental grouping plan for disuse muscle atrophy.

	Left Lower Limb	Right Lower Limb
Control group	No treatment (C-L)	No treatment (C-R)
2-week immobilization group	2-week immobilization and 2-week remobilization (2W-L)	No treatment (2W-R)
4-week immobilization group	4-week immobilization (4W-L)	No treatment (4W-R)

## Data Availability

The raw RNA-seq data reported in this study were submitted to the GSA database with accession number: CRA013067.

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
