# Peer review of "Transcriptome and Metabolome Profiling Provide New Insights into Disuse Muscle Atrophy in Chicken: The Potential Role of Fast-Twitch Muscle Fibers"

_ijms, 2024, doi:10.3390/ijms25063516_

Round 1
Reviewer 1 Report
Comments and Suggestions for Authors
The work done by Zipei Yao and cols entitled “Transcriptome and metabolome profiling provide new insights into disuse muscle atrophy in chicken: the potential role of fast-twitch muscle fibers” is focused on clarify the mechanisms involved in the development of disuse muscle atrophy in the limb of chicken. The animal model for disuse atrophy was the restricted movement of the left limb for 2 weeks + 2 weeks of recovery and 4 weeks of immobilization. The authors used the gastrocnemius muscle of the chicken to develop a significant amount of measurements and procedures including analysis of cross sectional area, measurement of several mRNA related to protein degradation pathways, analysis of fiber composition and mainly analysis of metabolome and transcriptome. Although the authors did a great effort to accomplish the aim of the study, it has a huge cavities that need to be addressed.
- Overall manuscript writing need to be revisited since it has several typos and writing errors.
- Intro needs to be rewritten, making clear the rationale for the study of skeletal muscle atrophy in chickens, why is this relevant in the area of knowledge further than just say it still remains unclear whether similar mechanisms apply to poultry.
- Grouping of animals is too confusing. In instance, what does it mean Negative Control? is there a positive control in the study? What treatment was applied to the animals?
-What is the rationale for the 2 week immobilization plus 2 week release to free movement? If the muscle moves freely, muscle structure eventually will recover.
-Not sure if right limb, which maintain the free movement, it is the most accurate control for its counterpart left limb, which is movement restricted, since those with free movement will be overload and as a consequence hypertrophic, which is evident in the figure 1 right panel and in figure 1 E and F in which is clear the frequency distribution % 2W-R is bigger as compared vs NC-R.
- Please clarify all abbreviations in the manuscript before use.
- Figure 1A H&E images are too small, making difficult to see an actual difference in fiber size between groups. Please provide images with higher magnification.
- How do the authors explain increased atrogin1 and MURF1 mRNA in 2W-R group?
- Results from transcriptome and metabolome are quite impressive; however, data description is also confusing. Please rewrite this section in a more fluid and coherent manner.
- Please include in the discussion section a paragraph highlighted the relevance of make this kind of studies in chicken
Comments on the Quality of English LanguageThe work done by Zipei Yao and cols entitled “Transcriptome and metabolome profiling provide new insights into disuse muscle atrophy in chicken: the potential role of fast-twitch muscle fibers” is focused on clarify the mechanisms involved in the development of disuse muscle atrophy in the limb of chicken. The animal model for disuse atrophy was the restricted movement of the left limb for 2 weeks + 2 weeks of recovery and 4 weeks of immobilization. The authors used the gastrocnemius muscle of the chicken to develop a significant amount of measurements and procedures including analysis of cross sectional area, measurement of several mRNA related to protein degradation pathways, analysis of fiber composition and mainly analysis of metabolome and transcriptome. Although the authors did a great effort to accomplish the aim of the study, it has a huge cavities that need to be addressed.
- Overall manuscript writing need to be revisited since it has several typos and writing errors.
- Intro needs to be rewritten, making clear the rationale for the study of skeletal muscle atrophy in chickens, why is this relevant in the area of knowledge further than just say it still remains unclear whether similar mechanisms apply to poultry.
- Grouping of animals is too confusing. In instance, what does it mean Negative Control? is there a positive control in the study? What treatment was applied to the animals?
-What is the rationale for the 2 week immobilization plus 2 week release to free movement? If the muscle moves freely, muscle structure eventually will recover.
-Not sure if right limb, which maintain the free movement, it is the most accurate control for its counterpart left limb, which is movement restricted, since those with free movement will be overload and as a consequence hypertrophic, which is evident in the figure 1 right panel and in figure 1 E and F in which is clear the frequency distribution % 2W-R is bigger as compared vs NC-R.
- Please clarify all abbreviations in the manuscript before use.
- Figure 1A H&E images are too small, making difficult to see an actual difference in fiber size between groups. Please provide images with higher magnification.
- How do the authors explain increased atrogin1 and MURF1 mRNA in 2W-R group?
- Results from transcriptome and metabolome are quite impressive; however, data description is also confusing. Please rewrite this section in a more fluid and coherent manner.
- Please include in the discussion section a paragraph highlighted the relevance of make this kind of studies in chicken
Author Response
Please see the attachment (Word file).

Reviewer 2 Report
Comments and Suggestions for Authors
Summary
This manuscript reports the phenotypic, transcriptomic, and metabolomic analyses of disuse muscle atrophy in chickens. The experiments seemed to be performed almost appropriately, but the bioinformatic analyses are arbitrary and not statistical as pointed out below. The significance of the genes, proteins, and/or pathways identified by the study is not fully guaranteed. Not only in bioinformatic analysis but also in wet experiments, the procedures are not adequate or not well described.
Comments
1. Section 4.9 and Figures 1H, 1I, 2B, 2C, 2D, 2F, and 2G: Comparisons between more than three groups must be analyzed using an appropriate multiple comparison test with analysis of variance (ANOVA). “ns” should be defined.
2. Section 4.4 and Figure 2A: What do the red dots mean? There is no information on the antibody used for immunofluorescence staining. I couldn’t understand how the authors identified MYH1B+ and MYH7+ myofibers in these images.
3. Figures 5A, 5B, 5E, and 5F: Label characters of these figures are lost. Use images with higher magnification. What do the red underlines in Figures 5E and 5F? At least, the class of “Fatty acid...” with the red underline is not statistically significant. In these analyses, the authors listed the “top 20” classes, but the list must be provided with the statistically appropriate threshold.
4. Figures 7E and 7F: Label characters are unreadable. As pointed out above, “top 20” is statistically meaningless.
5. Figures 8C and 8D: Use statistical threshold instead of “top 30”.
6. Figures 9A and 9B: Use statistical threshold instead of “top 10”.
Minor points
7. Line 75: “H&E” should be defined here.
8. Figure 2: Panel 2E is not in figure. Check the figure legend and main text.
9. Line 152: “FC” should be defined.
Author Response
Please see the attachment (Word file).

Round 2
Reviewer 2 Report
Comments and Suggestions for Authors
The authors sincerely revised the manuscript according to the reviewer's comments, especially in the description of the bioinformatics analysis.
Author Response
Dear Reviewers,
Thank you for acknowledging our efforts in revising the manuscript in line with your comments. We are glad that our revisions have met with your approval and we appreciate your positive feedback.
Should you have any further suggestions or require additional information, please do not hesitate to reach out. Your constructive feedback has been instrumental in improving the quality of our research, and we value your input.
Warm regards,
Zipei Yao